# Why a Naive Way to Combine Symbolic and Latent Knowledge Base Completion Works Surprisingly Well

**Christian Meilicke**                    CHRISTIAN@INFORMATIK.UNI-MANNHEIM.DE
**Patrick Betz**                          PATRICK@INFORMATIK.UNI-MANNHEIM.DE
**Heiner Stuckenschmidt**                 HEINER@INFORMATIK.UNI-MANNHEIM.DE
*University Mannheim, Data and Web Science Research Group*

## Abstract

We compare a rule-based approach for knowledge graph completion against current state-of-the-art, which is based on embeddings. Instead of focusing on aggregated metrics, we look at several examples that illustrate essential differences between symbolic and latent approaches. Based on our insights, we construct a simple method to combine the outcome of rule-based and latent approaches in a post-processing step. Our method improves the results constantly for each model and dataset used in our experiments.

## 1. Introduction and Motivation

In this paper we compare knowledge graph completion (KGC) methods, that are based on non-symbolic representations in terms of embeddings, against a symbolic approach that is based on rules. Given a knowledge graph $\mathcal{G}$, which is a set of triples, a KGC task is to predict the question mark in the incomplete triple $r(e, ?)$ or $r(?, e)$, where $r$ is a relation (binary predicate) used in $\mathcal{G}$ and $e$ is an entity described by (usually) several triples in $\mathcal{G}$. KGC methods solve this task by generating a ranking of possible candidates, which are evaluated by metrics that focus on the rank of the correct prediction within this ranking.

A first famous model for solving the KGC problem via a representation in an embeddings space is known as TransE [Bordes et al., 2013]. The basic idea of such a model, termed knowledge graph embedding (KGE) model in the following, is to first randomly map the entities and relations from $\mathcal{G}$ to a multidimensional space, e.g., to $\mathbb{R}^n$. Then the triples from $\mathcal{G}$ are used to set up a large optimization problem that is solved via gradient descent or a similar algorithm. How to transfer a triple into a part of the objective function is usually the defining characteristic of the model. By doing this, the final embedding becomes an alternative representation of the knowledge graph and it can be used to answer the completion task. Contrary to this, a rule-based approach searches for regularities in the graph and expresses them usually in the form of horn clauses. These rules are then used to propose candidates for solving the completion task.

Within this paper we try to understand the essential differences between both families of approaches. We first tackle this problem by analysing the rankings for several concrete completion tasks. This analysis results into several insights that motivates a specific method to combine both types of approaches. This aggregation method improves each of the incoming rankings, no matter whether rule or embeddings-based input has been better. As an additional benefit our method does not loose the explanatory power of a rule-based approach but integrates it into the combined approach.

## 2. Related Work

In [Zhang et al., 2021] the authors are concerned with different KGC methods with a focus on symbolic reasoning, neural reasoning techniques, and everything in between. However, they discuss differences from a rather abstract level and try to give a comprehensive overview without any connection to their empirical analysis. Symbolic approaches and latent approaches that are based on a projection to an embedding space have already been compared in an experimental study in [Meilicke et al., 2018], where the authors identify types of completion tasks for which there are differences between these approaches. However, within the last three years significant improvements have been made. For example, in the 2018 publication the authors reported about top hits@10 scores for the FB237 dataset in the range 0.4 to 0.43, while in [Ruffinelli et al., 2020] and [Rossi et al., 2021] scores between 0.5 and 0.55 are reported. This makes insights that follow from these experiments to some degree unreliable, as it might be the case many flaws are meanwhile fixed. Note also that our main interest is not to find types of completion tasks where rules are superior or vice versa but to understand at least partially why this is the case.

In [Rossi et al., 2021] the authors compare a large set of embeddings based methods and included the rule-based approach AnyBURL [Meilicke et al., 2019] as a baseline. This paper describes an extensive experimental study and comes to the conclusion that AnyBURL is an efficient alternative to non-symbolic methods. To our best knowledge, AnyBURL is the only symbolic approach that has proven to achieve performances on par with the current KGE state-of-the-art and therefore we choose AnyBURL as the representative for rule-based approaches within our experiments.

In the second half of this paper we combine the results of rule and embedding based approaches. The core principle of our strategy, combining multiple models into an ensemble, has been studied in [Wang et al., 2018] and in [Meilicke et al., 2018]. Although the approach in this paper can be seen as an extension, it differs fundamentally from a typical ensemble. As we discuss in Section 4, we exclusively allow for candidates which are within the language bias of the rule based approach by only considering its top-k candidates.

Our combination strategy ensures that the rule based method and the KGE model operate independently and are able to focus on their strengths before the results are aggregated to achieve an overall improvement. There are several papers that propose, on the contrary, to tightly integrate rule learning and embedding based approaches. Examples can be found in KALE [Guo et al., 2016] and RUGE [Guo et al., 2018].[1] However, approaches that belong to this category are mostly restricted to a type of rule which does not allow for constants.

More recently differentiable rule learning has been proposed [Rocktäschel and Riedel, 2017, Yang et al., 2017, Sadeghian et al., 2019, Minervini et al., 2020]. These approaches are not the focus of this work. Nevertheless, the aforementioned papers either have not been applied to the common KGC datasets or the results in this paper are better.

---

1. A comparison with these models is possible via the performance achieved on the FB15K dataset. Here the base version of AnyBURL alone performs significantly better.

## 3. Rule-Based Knowledge Base Completion with AnyBURL

We abstain from a detailed description of the AnyBURL algorithm and refer the reader to [Meilicke et al., 2019, 2020]. Nevertheless, for the purpose of this work it is important to know what kind of rules AnyBURL learns and how they are applied to create the final rankings.

### 3.1 Language Bias

We give concrete examples for the most important rule types within this section. There are two additional rule types that are mainly used for filling up the rankings which are explained in the Appendix in Section C. Each of the supported rules is a horn rule and can be written as $h \leftarrow b$. We call $h$ the head of the rule and $b$, which is a conjunction of atoms, the body of the rule.

The first and probably most prominent type of rules are called closed connected rules in [Galárraga et al., 2013] or cyclic rules according to [Meilicke et al., 2019]. The attribute cyclic refers to the head variables $X$ and $Y$ being directly connected in the head of the rule, while there is an alternative path expressed in the body of the rule. These rules do not contain constants but variables only. Here are some examples.

$$hypernym(X, Y) \leftarrow hyponym(Y, X) \tag{1}$$
$$contains(X, Y) \leftarrow contains(X, A), adjoins(A, Y) \tag{2}$$
$$contains(X, Y) \leftarrow administrative\_parent(A, X), adjoins(A, B), capital(B, Y) \tag{3}$$

The first rule expresses that the *hypernym* relation is inverse to the *hyponym* relation. Rule 2 expresses the transitivity of the contains relation. Rule 3 is a rather complex rule that expresses roughly that $X$ contains $Y$ if $X$ is an administrative parent location of $A$, $A$ shares a border with $B$ and $B$ has capital $Y$. Note that this rule is an example of a rule that creates highly ranked correct answers in our experiments.

Another important type of rules are acyclic rules with only one variable, that have a constant in the head and a (in most cases different) constant in the body. AnyBURL is restricted in its default setting, which we used in our experiments, to learn rules of this type with only one body atom. Here are two examples.

$$citizen(X, UK) \leftarrow bornIn(X, London) \tag{4}$$
$$contains(Australia, Y) \leftarrow contains(Victoria, Y) \tag{5}$$

Rule 4 expresses that someone born in London is (probably) a citizen of the UK. Rule 5 says that locations contained in Victoria are also contained in Australia. The vast majority of the mined rules are longer cyclic rules (as Rule 3) and acyclic rules similar to the ones that we just presented. A regularity that cannot be expressed in terms of these rule types is completely invisible to AnyBURL.

### 3.2 Applying Rules

To compute a prediction for a given completion task $r(a, ?)$ AnyBURL applies all rules that might create a triple $r(a, c)$ where $c$ is the predicted candidate. This is done by grounding

the relevant rules against the training set, i.e., by replacing the variables with constants (in other words: entities) such that the resulting body atoms are triples contained in the training set. Following this procedure[2] it will happen quite often that a candidate is predicted by several rules. In this case the maximum of the confidences of these rules is associated as confidence of predicting $c$. If there are several candidates $c$ and $c'$ that are predicted with the same confidence, they are ordered in the ranking according to the confidence of the second best rule, (if this confidence is also the same, the third-best counts, and so on). Two candidates that cannot be distinguished due to the fact that they are predicted by a set of rules that have exactly the same confidences, are ranked randomly.

We call this aggregation method in the following maximum-aggregation. In [Meilicke et al., 2020] the authors also reported about a noisy-or-aggregation where a candidate predicted by a set of rules will have a higher confidence than a candidate predicted by a subset. It turned out that this method performed worse on all datasets compared to the simpler maximum-aggregation. This was especially caused by the problem that the method cannot discriminate between redundant rules that fire for the same reason and rules that capture different aspects. Opposed to a symbolic approach, that requires to explicitly define an aggregation type, KGE approaches have an implicit aggregation technique that is based on the fact that each triple is part of a comprehensive objective function. This is another advantage that KGE models might have compared to rule-based approaches as long as they are based on a rather simple aggregation method.

## 4. Case By Case Analysis

It is not easy to distinguish accidental differences, that might be caused by the stochastic nature of the approaches, from systematic differences. We first propose a method to spot essential differences before we discuss what we found with this method. We report occasionally about datasets and KGE models that are first introduced in Section 6.1.

### 4.1 Spotting Essential Differences

Let $\mathcal{R}$ be a candidate ranking for a completion task $r(e, ?)$. A candidate ranking is a total order over the entities in the given knowledge graph. We use $\mathcal{R}[n]$ to denote the entity ranked at position $n$ in $\mathcal{R}$ and $\mathcal{R}[c]^{\#}$ to denote the ranking position of a candidate $c$. Given the completion task $r(e, ?)$, let $\mathcal{A}$ denote the AnyBURL ranking for $r(e, ?)$ and let $\mathcal{E}$ be a ranking for $r(e, ?)$ generated by a KGE model. We use $conf(c, r(e, ?))$ to denote the confidence that AnyBURL assigns to $c$ with respect to being the answer to $r(e, ?)$. Then

$$\Psi(c, \mathcal{E}, r(e, ?)) = \frac{conf(c, r(e, ?))}{conf(\mathcal{A}[\mathcal{E}[c]^{\#}], r(e, ?))}$$

denotes the anomaly degree of ranking $c$ in $\mathcal{E}$ for $r(e, ?)$ from the perspective of AnyBURL. The definition is based on the idea to compare the confidence that AnyBURL assigns to $c$ to the confidence that AnyBURL assigns within its own ranking to the entity ranked at the position where $c$ is ranked in the KGE ranking. A score of around 1 means that there is no

---

2. The actual AnyBURL algorithm is a bit more complicated, however, its final result corresponds to the result of the procedure described here.

| | AnyBURL | | ComplEx | | RESCAL | |
|---|---|---|---|---|---|---|
| Rank | Candidate | Confidence | Candidate | Score | Candidate | Score |
| #1 | **Australia** | 0.659 | South Australia | 11.780 | New South Wales | 1.683 |
| #2 | USA | 0.032 | Queensland | 11.226 | **Australia** | 1.385 |
| #3 | Canberra | 0.026 | New South Wales | 10.323 | New Zealand | 1.221 |
| #4 | South Australia | 0.017 | Western Australia | 9.515 | South Australia | 0.276 |
| #5 | New South Wales | 0.017 | Tasmania | 8.676 | Queensland | 0.249 |
| #6 | Western Australia | 0.017 | Victoria | 8.539 | United Kingdom | 0.074 |
| #7 | Queensland | 0.017 | **Australia** | 8.338 | England | -0.064 |

Table 1: Rankings for the completion task *contains(?, Darwin)*

anomaly, high scores mean that AnyBURL is highly confident that $c$ is ranked to low, and low scores mean that $c$ ranked to high.

One might argue that it would have been better to base the definition of $\Psi$ on a direct comparison of the ranking positions. If we have a completion task as *locatedIn(?, UK)* a city as Bristol might be ranked by AnyBURL on position #12 while it might be ranked on #77 by a KGE model. However, the AnyBURL confidences between rank #10 and #80 might be very close. Our definition of $\Psi$ takes this into account and would yield a score around 1, while we would get a high score if we would base the score on ranking positions.

## 4.2 Selected Examples

We first look at three predictions $c$ with $\Psi(c, \mathcal{E}, r(e, ?)) < 0.2$ that are ranked above the correct hit. In the last paragraph we talk about the opposite, in particular we report about a relation where we spotted high $\Psi$ scores in general. We sometimes mark assertions with '(in test/training)'. This means that the corresponding triple can be found in the test set (or in the training set). Additional examples are discussed in the appendix in Section B.

**Where is the city Darwin**   The city Darwin is a city in Australia (in test). Darwin is located in the Northern Territory (in training). The training set contains another triple that states that Darwin is also the capital of the Northern Territory. The training set states also that Australia contains the Northern Territory. We are now concerned with the completion task *contains(?, Darwin)*. As shown in Table 1 AnyBURL puts Australia first with all other alternatives having a very low confidence, while ComplEx ranks each Australian territory first before Australia appears at #7. RESCAL puts it on #2, however, its scores are similar to the scores of other alternatives that are clearly wrong.

AnyBURL ranks *Australia* first due to Rule (3), a complex rule with three body atoms, presented in Section 3.1. The rule that expresses the transitivity of the contains relation (a rule with two atoms in the body) would also be sufficient to put Australia on top with a confidence of 0.273. It seems that both regularities have a rather limited impact on the KGE rankings. The KGE ranking might be affected by the fact that the other Australian territories are very similar to the Northern Territory (all are contained in Australia, some share borders with each other, ect.). The KGE ranking can be explained by a substitution of Northern Territory in *contains(Northern Territory, Darwin)* by very similar entities.

**Metropolis** The completion task *festivals(Metropolis,?)* asks at which festival the movie Metropolis has been shown. The correct answer is the 39th Berlin International Film Festival. AnyBURL places the correct answer at #18 with a low confidence of 0.016, ComplEx ranks it at #25. This difference is not significant. More interesting are the candidates that can be found in the complete ranking. The first 23 AnyBURL ranks are filled with film festivals, followed by a mixed list that contains both festivals and cities (the relation *festivals* has sometimes been used to express that a movie has been shown in a certain city). All candidates have a low confidence. Still, they are to a certain degree meaningful up to (at least) position #36. In the ComplEx ranking starting from position #23 weird candidates show up: USA (#23), natural death cause (#24), Chris Parnell (#31), electric guitar (#33). What is interesting with respect to these candidates is not that they exists somewhere in the ranking but that they are ranked above some meaningful candidates without obvious reasons. It seems that the signals for the other meaningful candidates are too weak to enforce a meaningful order.

**Michael Fisher works for King's College** This example is concerned with the completion task *employer(fisher,?)*. Michael Fisher has been a mathematician and physicist, who studied at the Kings's College (training set) and worked as an employee at the Kings's College (test set). Another relatively important triple given in the training set states that Michael Fisher worked also for the Leiden University. AnyBURL ranks the correct answer, King's College, at position #10 only, while ComplEx and most of the other KGE models rank King's College at #1 or at least among the top-5. When analysing the task from the perspective of AnyBURL we detected the following two rules, that result into the correct prediction. The rules are shown on the left, the triples that makes the rule fire are shown on the right.

$$employer(X,Y) \leftarrow studiedAt(X,Y) \qquad studiedAt(fisher, kingscoll) \qquad (6)$$
$$employer(X, kingscoll) \leftarrow employer(X, leiden) \qquad employer(fisher, leiden) \qquad (7)$$

Both rules have a confidence of $\approx 8\%$. There are several rules with higher confidences (up to 15%) that generate the candidates that are ranked above King's College. All other rules, which allow to predict King's College, have a confidence lower than 2%. As both rules have nearly the same confidence, the maximum aggregation in AnyBURL will score King's College with (nearly) the same score no matter if we have both or only one of the two triples. If we remove both, King's college falls out of the top-50 ranking.

This example allows us to shed light on two question: (1) Are the triples that determine the AnyBURL ranking the same triples that determines the KGE behaviour? (2) To what extent can the KGE results be explained as a cumulative aggregation of these triples (and the rules that they fire)? For this purpose we have executed ComplEx and HittER* on the original dataset, on the variant where we removed ($t_1$) *studiedAt(fisher,kingscoll)*, on the variant where we removed ($t_2$) *employer(fisher,leiden)*, and on the variant were we removed both. As KGE models can vary a lot between different runs[3], we conducted six runs,

---

3. It is sometimes argued that MRR scores are relatively stable between different runs. This observation is not an objection to our claim as the MRR sums up scores from many completion tasks that might differ on the fine-grained level.

based on a different random initialisation, for each dataset variant. Results are reported in Table 2.

| | ComplEx | | HittER* | |
|---|---|---|---|---|
| | Avg.Rank (Std) | Avg.Score | Avg.Rank (Std) | Avg.Score |
| All triples | 1.5 (0.83) | 5.53 | 1.66 (0.81) | 5.16 |
| Without $t_1$ | 2.83 (1.32) | 4.95 | 10.8 (19.85) | 4.39 |
| Without $t_2$ | 31 (24.43) | 3.19 | 44.16 (25.37) | 1.30 |
| Without $t_1$ and $t_2$ | 68 (53.28) | 2.28 | 97 (74.63) | -0.17 |

Table 2: The impact of removing triples w.r.t a specific completion task.

Contrary to the maximum aggregation of AnyBURL, both KGE models aggregate the evidence that lies within these triples in a beneficial way. When suppressing $t_1$ the average rank of the correct hit falls slightly for ComplEx and significantly for HittER*. The impact of removing $t_2$ is similar, yet stronger. If both triples are removed, King's College drops to a rank below #50. This shows that the triples that are relevant for the AnyBURL rankings have also a significant impact on the rankings of ComplEx and HittER*.

This kind of aggregation seems to work in general better for the tail predictions related to the *employer*-relation. We computed the MRR for these completion tasks only. The KGE models achieved a score between 0.38 and 0.43, while the MRR of AnyBURL is 0.34 only. With only few exception all rules learned for this relation have a confidence less then 0.4. We further looked at several randomly selected examples and for most of them several low-confident rules fired similar to the example we presented above.

## 5. Aggregating Rankings

In the following we propose an approach to combine the rankings generated by a rule-based approach and the rankings generated by a KGE model. Motivated by the analysis in the previous section, we try to achieve the following goals:

- If something appears in a KGE ranking (1) which is an artefact of an random initialisation (e.g., *Metropolis*) or (2) which appears there due to a similarity consideration not backed by a regularity (e.g., *Darwin*), it should not appear in the final ranking.

- Relations for which KGE methods work better (e.g., *employer*) and relations for which rules work better should be treated differently. The method should choose a weighting between KGE and rules that fits best to the relation and direction (head vs. tail).

The first requirement can be fulfilled by suppressing any (top-ranked) KGE prediction that is not at all predicted (or predicted with a confidence close to 0) by a rule based approach. Thus, the approach that we describe in the next section is restricted to the top-k ranking of AnyBURL and uses the KGE scores only as an additional information to change the position in the ranking of AnyBURL. This will directly filter out any predictions for which a rule-based approach does not see any evidence. It is a rather risky approach, as it is build on the assumption that everything visible to KGE lies within the language bias

of AnyBURL. In case our approach works well, this shows that the vast majority of KGE predictions can be backed up by a symbolic explanation.

The second requirement can easily be implemented by using the validation set to determine for which relations KGE models support better results and for which AnyBURL creates better rankings. As it might be the case that there is a difference between head and tail-predictions within the relation, we have to distinguish not only between different relations but have to take the direction of the prediction additionally into account.

For each relation $r$ we first collect all triples from the validation set that use $r$. We refer to this subset as $V(r)$. Now we search over possible values for an aggregation parameter $\beta_{r,ht}$ where $r$ denotes the relation and $ht$ determines whether we deal with head or tail predictions. We iterate over all head (or tail) completion tasks $r(?, e)$ (or $r(e, ?)$) resulting from the triples in $V(r)$. For each prediction task $r(e, ?)$ we compute an aggregated score $score_{agg}$ for each candidate in $\mathcal{A}_k$, where $\mathcal{A}_k$ denotes the top-k candidate ranking created by AnyBURL. We use the following formula where $score_{norm}(c, r(?, e))$ is the normalized score that the KGE model assigned to $c$ in the context of $r(?, e)$. We explain in the appendix in Section E how we normalize the KGE score.

$$score_{agg}(c, r(?, e)) = \beta_{r,ht} * conf(c, r(?, e)) + (1 - \beta_{r,ht}) * score_{norm}(c, r(?, e))$$

The aggregated score is a linear combination of AnyBURL and normalized KGE score, where $\beta_{r,ht}$ determines the weighting. Based on the aggregated scores, we create a reordered ranking. Once we computed all aggregated rankings for the tail (or head) predictions in $V(r)$ for a specific $\beta_{r,ht}$, we compute the MRR for these rankings. We search for the best parameter $\beta_{r,ht}$ for each relation and direction (head vs. tail prediction) via a grid search.

## 6. Experiments

We first explain the settings and datasets that we used in our experiments in Section 6.1, followed by a presentation of the most important results in Section 6.2 and further ablation experiments in Section 6.3.

### 6.1 Settings

We evaluate our approach on FB237 [Toutanova and Chen, 2015] (also called FB15k-237 or FB15KSelected) and WNRR [Dettmers et al., 2018], which are frequently used in the literature and have been created to overcome leakage and redundancy problems of FB15K and WN18 [Bordes et al., 2013], respectively. Furthermore, we use the CoDEx benchmark [Safavi and Koutra, 2020] which includes three knowledge graphs in varying sizes designed with the goal to be more difficult than previously published datasets [Safavi and Koutra, 2020]. Summary statistics for the datasets can be found in Table 8 in the appendix.

In regard to the KGE models, we use the libKGE library [Broscheit et al., 2020] which focuses on reproducibility and has shown to produce state-of-the-art results. We include TransE [Bordes et al., 2013], RESCAL [Nickel et al., 2011], DistMult [Yang et al., 2015], ComplEx [Trouillon et al., 2016], ConvE [Dettmers et al., 2018], and TuckER [Balažević et al., 2019]. For these models, we use the pretrained embeddings from libKGE when available for the respective datasets. Due to its very good results, we use additionally the transformer implementation of libKGE which is based on the HittER no-context model [Chen

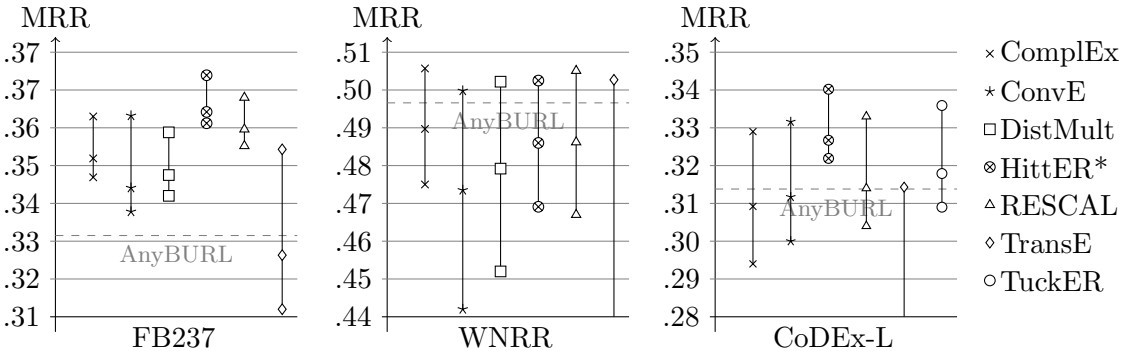

Figure 1: MRR of AnyBURL, KGE (bottom), filtered (mid) and aggregated results (top).

et al., 2020]. We refer to this implementation, which is mentioned as transformer in libKGE, as HittER* in the following.[4] More details can be found in the supplementary material.

With respect to the aggregation method described in Section 5 , we set $k$=100. We search the best $\beta_{r,ht}$ value over each multiple of 0.05 in the range $[0, 1]$. We use the evaluation protocol that has first been proposed in [Bordes et al., 2013]. In particular, we report hits@1, hits@10 and MRR (mean reciprocal rank). In the following we always present the filtered variants of these measures without adding the adjective 'filtered'.[5]

## 6.2 Main Results

In particular, we compare the MRR scores for KGE, AnyBURL and the aggregated results in Figure 1 for WNRR, FB237, and CoDEx-L (all CoDEx variants can be found in the appendix in Figure 2). Detailed numbers can be found in the appendix in Table 4. We have depicted a vertical line for each KGE model. The lower endpoint refers to the MRR of the model itself, and the upper endpoint refers to the MRR that we measured after applying our aggregation method. The longer this vertical line, the stronger is the positive impact of our approach. We explain the marker in the middle of each vertical line in Section 6.3. We have depicted the AnyBURL results as a horizontal dashed line.

We need to point to some general differences first of all. The WNRR datasets is a dataset where AnyBURL clearly outperforms each of the KGE models in terms of MRR. For FB237 the opposite is the case with the exception of TransE. The largest version of the CoDEx datasets is somewhere in between as there are some KGE models that are better and some that are worse than AnyBURL. We observe a similar trend across all datasets and KGE models. Each KGE model is improved by at least one percentage point on each dataset. On average the improvement is 2.6 percentage points excluding TransE and 4.2 including TransE. The same holds from the perspective of AnyBURL. Even in the case of

---

4. For HittER*, the libKGE developers provided us with hyperparameter configurations for the FB237 dataset which we use for training the model. For the remaining datasets, we run the hyperparameter search provided by libKGE where the search space is centered around the FB237 configuration.

5. Our MRR is based on the top-100 rankings and any correct candidate ranked below is not taken into account, which means that the reported MRR of our aggregation method is slightly worse (at most by -0.001) compared to the standard MRR. To avoid any unfair comparison we present the standard MRR scores for the non-aggregated KGE models, which are based on the complete ranking.

WNRR, each of the models, inlcuding those that do not perform well, help to improve the results of AnyBURL by at least 0.5 percentage points.

The models that do not perform well (e.g., TransE on FB237, RESCAL on WNRR, TransE on CoDEx-M) are improved significantly. As a result the gap between the results that we have after the aggregation do not differ much between different models anymore. This in particular important, as we are talking about models that differ a lot in terms of computational effort required to find a good hyperparameter setting.

One of the best models in our evaluation is HittER*. Here we observe an improvement of $\approx 2$ percentage points for CoDEx-L, an improvement of more than 3 percentage points on WNRR and and improvement of roughly 1.5 percentage points on FB237. Especially the FB237 result is interesting, as it shows that the aggregation with AnyBURL can improve a top-score even though the AnyBURL result itself is significantly worse. Please also note that we were able to improve the HittER* results on FB237 and WNRR up to the performance of the full-fledged HittER variant, that has been claimed to be state-of-the-art in [Chen et al., 2020]. This is in particular interesting as our approach requires significantly less computational resources to achieve the same results. Results for the full-fledged HittER variant for CoDEx-L are not available, and it would be extremly costly to generate these results.

### 6.3 Ablation Study

To better understand what causes these results, we conducted experiments where we analyzed the impact of the $beta_{r,ht}$ scores. First, we fixed the $beta_{r,ht}$ score to 0. This means that the rankings are completely determined by the KGE scores, while the candidates that are ranked are those provided by AnyBURL. By doing this, we are able to use our method as a filtering technique that suppresses any artifact from the random initialisation and anything based on a similarity consideration not backed by a regularity that corresponds to a rule learned by AnyBURL. It turned out that the resulting MRR is always between the original KGE score and the MRR of the aggregated results. We depict this score as a mark within the vertical lines in Figure 1. Detailed results of these experiments can be found in Table 5. Between 1/4 and 1/2 of the positive impact can be explained by filtering out what is not predicted by AnyBURL. This result implies also that the KGE models are not capable of detecting anything that cannot be described in terms of the rules we presented in Section 3.1.

We further explore the behavior for different values of $\beta_{r,ht}$. We show results of experiments for HittER* and ComplEx on CoDEx-L in Table 3. The $\beta_{r,ht}$=0.0 setting corresponds to using AnyBURL as a filter as explained above. Setting $\beta_{r,ht}$=1.0 corresponds to the original AnyBURL scores. In the line that shows results for $\beta_{r,ht}$=0.5, both members of the ensemble have an equal and fixed weight. In the last row we show the results of our approach in its standard setting from Figure 1 where we allow to select the best $\beta_{r,ht}$ from $\{0, 0.05, \ldots, 1.0\}$ based on the best MRRs on the validation set. In the row above we restrict the search space to $\beta_{r,ht} \in \{0, 1\}$.

The results show that an equally balanced approach can already improve the performance. However, learning an optimal $\beta_{r,ht}$ against the validation set yields further improvements. This holds especially for HittER*, which is the best model in our experiments.

|  | ComplEx | | | HittER* | | |
|---|---|---|---|---|---|---|
|  | h@1 | h@10 | MRR | h@1 | h@10 | MRR |
| Original numbers | 0.237 | 0.400 | 0.294 | 0.257 | 0.447 | 0.322 |
| $\beta_{r,ht} = 0.0$ (filter only) | 0.247 | 0.430 | 0.309 | 0.262 | 0.453 | 0.327 |
| $\beta_{r,ht} = 0.5$ (equally balanced) | 0.266 | 0.446 | 0.326 | 0.269 | 0.454 | 0.331 |
| $\beta_{r,ht} = 1.0$ (AnyBURL) | 0.256 | 0.427 | 0.314 | 0.256 | 0.427 | 0.314 |
| $\beta_{r,ht} \in \{0, 1\}$ | 0.258 | 0.437 | 0.318 | 0.270 | 0.456 | 0.333 |
| $\beta_{r,ht} \in \{0, 0.05, \dots, 1.0\}$ | 0.268 | 0.447 | 0.329 | 0.277 | 0.463 | 0.340 |

Table 3: Exemplary results for different $beta_{r,ht}$ setting on CoDEx-L.

Moreover, learning a relation and direction specific weighting is superior to selecting one of the two approaches for each relation/direction, which is reflected by the $\beta_{r,ht} \in \{0, 1\}$ setting which is 0.7 and 1.1 percentage points worse compared to the main results.

In Tables 9 to 12 in the Appendix, for the 10 most frequent relations of CoDEx-L we present the $\beta_{r,ht}$ values that resulted in the best MRR scores against the validation set in regard to the last two settings of Table 3. A $\beta_{r,ht}$ of 0.0 means that the KGE model determines the ranking, while a value of 1.0 means that the ranking is completely determined by AnyBURL. The $\beta_{r,ht}$ scores vary between the values of the respective search spaces, indicating that it a weighting where none of the two ensemble members is ignored is beneficial for most relations.

## 7. Conclusions

Instead of presenting a sophisticated new knowledge base completion method, in this work, we tried to understand advantages and disadvantages of KGE models and rule-based approaches by analysing the rankings that they generate. Our means to achieve this goal was the use of the explanatory power of a rule based approach. Thus, we were able to spot and understand several examples of interesting predictions that revealed some essential differences. As a consequence of our understanding, we developed an approach that increases the quality of an already top-performing KGE approach consistently by 1 to 3 percentage points in terms of the MRR. While the prediction quality of our aggregation method is in itself a valuable results, it is more important to understand what follow from these results:

- KGE models are good in combining/aggregating different signals.

- KGE models can suffer from relicts of the random initialization.

- KGE scores are affected by similarity considerations that are sometimes unreasonable.

- Rule based approaches are better in detecting signals that can be explained in terms of (relatively) long rules.

- KGE models remain within the language scope described in Section 3.1, which means that we can use the rules of AnyBURL to explain the predictions of the KGE model.

These insights explain why a naive way to combine symbolic and latent knowledge graph completion techniques works surprisingly well.

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

## Appendix A. Detailed Results

Table 4 shows detailed results in terms of filtered hits@1, hits@10 and MRR (based on the top-100 predictions only) for all combinations of KGE models and datasets.

| | Approach | Rules/KGE Results | | | Aggregated Results | | | Improvements | | |
|---|---|---|---|---|---|---|---|---|---|---|
| | | h@1 | h@10 | MRR | h@1 | h@10 | MRR | h@1 | h@10 | MRR |
| FB237 | AnyBURL | .246 | .506 | .332 | | | | | | |
| | ComplEx | .253 | .536 | .347 | .270 | .551 | .363 | +.017 | +.015 | +.016 |
| | ConvE | .248 | .521 | .338 | .273 | .548 | .363 | +.024 | +.027 | +.025 |
| | DistMult | .249 | .531 | .342 | .266 | .548 | .359 | +.017 | +.017 | +.017 |
| | HittER* | .268 | .549 | .361 | .283 | .56 | .374 | +.015 | +.01 | +.013 |
| | RESCAL | .263 | .541 | .355 | .275 | .555 | .368 | +.012 | +.015 | +.013 |
| | TransE | .221 | .497 | .312 | .264 | .536 | .354 | +.043 | +.039 | +.042 |
| WN18RR | AnyBURL | .457 | .572 | .497 | | | | | | |
| | ComplEx | .438 | .547 | .475 | .464 | .590 | .506 | +.026 | +.042 | +.031 |
| | ConvE | .411 | .505 | .442 | .459 | .580 | .500 | +.049 | +.075 | +.058 |
| | DistMult | .414 | .531 | .452 | .460 | .583 | .502 | +.047 | +.053 | +.050 |
| | HittER* | .437 | .531 | .469 | .463 | .583 | .503 | +.026 | +.052 | +.033 |
| | RESCAL | .439 | .517 | .467 | .465 | .582 | .505 | +.026 | +.065 | +.038 |
| | TransE | .053 | .520 | .228 | .458 | .591 | .503 | +.405 | +.071 | +.275 |
| CoDEx-S | AnyBURL | .341 | .622 | .436 | | | | | | |
| | ComplEx | .372 | .646 | .465 | .373 | .655 | .467 | +.001 | +.010 | +.002 |
| | ConvE | .343 | .635 | .444 | .361 | .649 | .457 | +.019 | +.013 | +.013 |
| | HittER* | .353 | .641 | .453 | .376 | .654 | .468 | +.023 | +.013 | +.015 |
| | RESCAL | .294 | .623 | .404 | .370 | .654 | .466 | +.077 | +.032 | +.062 |
| | TransE | .219 | .634 | .354 | .361 | .660 | .459 | +.143 | +.026 | +.105 |
| | TuckER | .339 | .638 | .444 | .375 | .652 | .468 | +.035 | +.015 | +.024 |
| CoDEx-M | AnyBURL | .247 | .45 | .316 | | | | | | |
| | ComplEx | .262 | .476 | .337 | .277 | .492 | .349 | +.015 | +.017 | +.012 |
| | ConvE | .239 | .464 | .318 | .274 | .487 | .346 | +.035 | +.024 | +.028 |
| | HittER* | .262 | .486 | .339 | .289 | .498 | .359 | +.027 | +.012 | +.021 |
| | RESCAL | .244 | .456 | .317 | .273 | .484 | .344 | +.028 | +.028 | +.027 |
| | TransE | .223 | .454 | .303 | .266 | .480 | .340 | +.043 | +.026 | +.037 |
| | TuckER | .259 | .458 | .328 | .274 | .482 | .344 | +.015 | +.024 | +.016 |
| CoDEx-L | AnyBURL | .256 | .427 | .314 | | | | | | |
| | ComplEx | .237 | .400 | .294 | .268 | .447 | .329 | +.031 | +.047 | +.035 |
| | ConvE | .240 | .420 | .300 | .269 | .453 | .332 | +.029 | +.033 | +.032 |
| | HittER* | .257 | .447 | .322 | .277 | .463 | .340 | +.021 | +.016 | +.018 |
| | RESCAL | .242 | .419 | .304 | .273 | .451 | .333 | +.031 | +.032 | +.029 |
| | TransE | .116 | .317 | .187 | .255 | .428 | .314 | +.139 | +.111 | +.127 |
| | TuckER | .244 | .43 | .309 | .274 | .455 | .336 | +.030 | +.025 | +.027 |

Table 4: Detailed results.

Table 5 is structured similar to Table 4, however, this time the results are based using the AnyBURL ranking only for the purpose of filtering the KGE ranking. This means that we keep the order of the KGE ranking and suppress each candidate that is not within the top-100 ranking of AnyBURL. The empty positions in the ranking are filled up by the subsequent candidates.

| | Approach | Rules/KGE Results | | | Filtered Results | | | Improvements | | |
|---|---|---|---|---|---|---|---|---|---|---|
| | | h@1 | h@10 | MRR | h@1 | h@10 | MRR | h@1 | h@10 | MRR |
| **FB237** | AnyBURL | .246 | .506 | .332 | | | | | | |
| | ComplEx | .253 | .536 | .347 | .259 | .541 | .352 | +.006 | +.005 | +.005 |
| | ConvE | .248 | .521 | .338 | .254 | .528 | .344 | +.006 | +.008 | +.006 |
| | DistMult | .249 | .531 | .342 | .255 | .536 | .348 | +.006 | +.004 | +.005 |
| | HittER* | .268 | .549 | .361 | .273 | .552 | .364 | +.005 | +.002 | +.003 |
| | RESCAL | .263 | .541 | .355 | .268 | .546 | .36 | +.005 | +.006 | +.004 |
| | TransE | .221 | .497 | .312 | .237 | .509 | .326 | +.016 | +.012 | +.014 |
| **WN18RR** | AnyBURL | .457 | .572 | .497 | | | | | | |
| | ComplEx | .438 | .547 | .475 | .448 | .575 | .490 | +.010 | +.027 | +.015 |
| | ConvE | .411 | .505 | .442 | .434 | .551 | .474 | +.023 | +.047 | +.032 |
| | DistMult | .414 | .531 | .452 | .437 | .563 | .479 | +.023 | +.033 | +.027 |
| | HittER* | .437 | .531 | .469 | .447 | .566 | .486 | +.010 | +.035 | +.017 |
| | RESCAL | .439 | .517 | .467 | .449 | .557 | .486 | +.010 | +.04 | +.019 |
| | TransE | .053 | .520 | .228 | .337 | .570 | .422 | +.284 | +.049 | +.194 |
| **CoDEx-M** | AnyBURL | .247 | .45 | .316 | | | | | | |
| | ComplEx | .262 | .476 | .337 | .266 | .487 | .341 | +.003 | +.011 | +.004 |
| | ConvE | .239 | .464 | .318 | .248 | .475 | .325 | +.009 | +.011 | +.007 |
| | HittER* | .262 | .486 | .339 | .267 | .487 | .342 | +.006 | +.001 | +.003 |
| | RESCAL | .244 | .456 | .317 | .250 | .469 | .323 | +.006 | +.014 | +.006 |
| | TransE | .223 | .454 | .303 | .238 | .461 | .312 | +.015 | +.008 | +.009 |
| | TuckER | .259 | .458 | .328 | .262 | .471 | .333 | +.003 | +.013 | +.005 |
| **CoDEx-L** | AnyBURL | .256 | .427 | .314 | | | | | | |
| | ComplEx | .237 | .400 | .294 | .247 | .430 | .309 | .010 | .030 | .015 |
| | ConvE | .24 | .420 | .300 | .247 | .439 | .312 | .007 | .019 | .012 |
| | HittER* | .257 | .447 | .322 | .262 | .453 | .327 | .006 | .007 | .005 |
| | RESCAL | .242 | .419 | .304 | .252 | .436 | .314 | .010 | .017 | .010 |
| | TransE | .116 | .317 | .187 | .172 | .363 | .236 | .056 | .046 | .049 |
| | TuckER | .244 | .430 | .309 | .254 | .442 | .318 | .010 | .012 | .009 |

Table 5: Using the candidate ranking of AnyBURL for the purpose of filtering only.

In addition to the visualisation shown in the main paper, we prepared the same diagram for CoDEx-S, M and L in Figure 2. Note that the validation set of CoDEx-S is relatively small, which might explain why the method works not so well for some models.

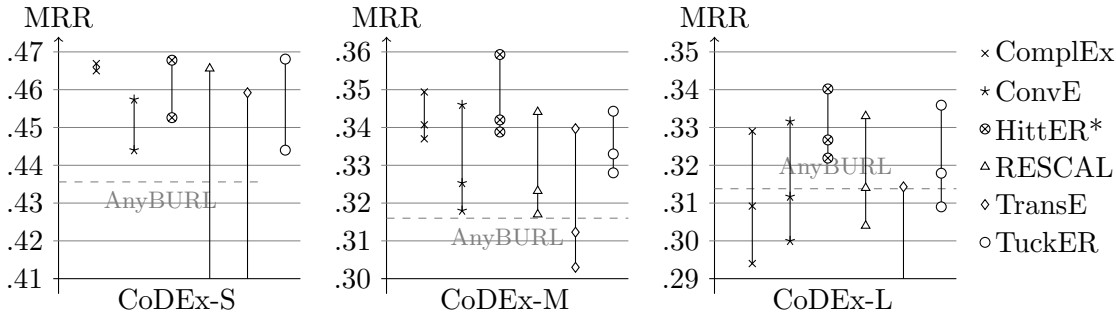

Figure 2: Comparing MRR of AnyBURL, embeddings-based models, and improved combined results on CODEX-S/M/L.

## Appendix B. Additional Examples

In the following we present two additional examples. The first example illustrates a behaviour similar to what we discussed with the *Darvin/Australia example*. The second example is related to a specific pattern. We have also developed a simple method to measure the impact of suppressing this pattern and report about its results.

**Who produced Tomb Raider**  We are now concerned with the completion task *producedBy(TombRaiderI,?)*. We use TombRaiderI/II to refer to the first and second movie from the series (in training is a prequel relation between these movies). The producer of both movies is Lawrence Gordon (in test for TombRaiderI; in training for TombRaiderII). Angeline Jolie acted in both movies as Lara Croft (both triples are available in training). It is also relevant that Angelina Jolie produced several other movies.

AnyBURL ranks the correct answer on position #1. This is caused by a rule with two atoms in the body that captures the regularity that the producer of a movie has produced the prequel of this movie with a probability of 0.445. ComplEx ranks Angelina Jolie first. What might be the reason for this result? It happens sometimes that a movie is produced by one of the persons acting in that movie. Moreover, the *producedBy* and the *actedIn* relations establish connections between the same types of entities. In that sense the embedding of these relations might be similar. Thus, as it is known that Angelina Jolie acted in TombRaider I, the triples *actedIn(TombRaider I, Angelina Jolie)* must have a high score, and thus, the triple *producedBy(TombRaider I, Angelina Jolie)* will also receive a relatively high score. On the other hand, the regularity that is captured by the rule that triggered AnyBURLs decision seems to affect the results only to a limited degree.

**Robert Schumann did not influence Robert Schumann**  A correct answer to the completion task *influencedBy(schumann,?)* is (beside others) the composer Felix Mendelsohn. However, ComplEx puts Robert Schumann himself on the first rank resulting in the triple *influencedBy(schumann, schumann)*. The correct answer (Mendelsohn) is ranked on position #9. Note that we observed many similar cases where a triple $r(a,a)$ was predicted, even though relation $r$ is, as it is the case for the *influencedBy* relation, irreflexive. While this sounds like a rather specific issue, it can again be explained by misleading similarity

|  | Rank | AnyBURL | | ComplEx | | RESCAL | |
|---|---|---|---|---|---|---|---|
|  |  | Candidate | Confidence | Candidate | Score | Candidate | Score |
| (A) | #1 | **Lawrence Gordon** | 0.445 | Angelina Jolie | 10.064 | Avi Arad | 4.025 |
|  | #2 | Michael G. Wilson | 0.222 | Steven Spielberg | 9.134 | Kathleen Kennedy | 3.833 |
|  | #3 | Steven Spielberg | 0.2 | **Lawrence Gordon** | 9.112 | **Lawrence Gordon** | 3.077 |
| (B) | #1 | Arthur Schopenhauer | 0.36 | Robert Schumann | 9.062 | Arnold Schoenberg | 2.571 |
|  | #2 | Victor Hugo | 0.25 | Bach | 8.232 | Thomas Mann | 2.514 |
|  | #4 | Spinoza | 0.231 | Schopenhauer | 8.116 | **Mendelssohn** | 2.061 |
|  | #9 | Jean-Jacques Rousseau | 0.153 | **Mendelssohn** | 7.517 | Sigmund Freud | 1.684 |
|  | #28 | **Mendelssohn** | 0.07 | Aleksandr Pushkin | 5.853 | Friedrich Hayek | 0.427 |

Table 6:  Ranking results for the completion tasks (A) *producedBy(TombRaiderI,?)* and (B) *influencedBy(schumann,?)*.

consideration. There several triples given in the training set, which tell us about people that have influenced Robert Schumann. Robert Schumann seems to be very similar to the average influencer of Schumann. For that reason he is proposed as a solution to the completion task. Some predictions for AnyBURL, ComplEx and RESCAL are shown in Table 6.

Note also that AnyBURL puts the correct answer to a rather low ranking position. However, that is not the point here. The main point here is related to the fact that an alternative that is clearly wrong, given the fact that we never observe *influencedBy(c,c)* for any constant $c$ in the training set, is placed at a top position by the KGE approach due to misleading similarity considerations.

We applied the following strategy to remove erroneous self-referential predictions from the ranking. For each relation $r$ We checked in the training set, whether there is a triples $r(c, c)$. If such a triples does not exist, we marked the relation as irreflexive. For each irreflexive relation $r$, given the prediction task $r(c, ?)$ or $r(?, c)$ we removed $c$ from the candidate ranking. Note that this filtering technique removes exactly one candidate or no candidates from a given ranking. The results of the approach are depicted in Table 7.

| | Approach | Rules/KGE Results | | | Filtered Results | | | Improvements | | |
|---|---|---|---|---|---|---|---|---|---|---|
| | | h@1 | h@10 | MRR | h@1 | h@10 | MRR | h@1 | h@10 | MRR |
| FB237 | ComplEx | .253 | .536 | .347 | .256 | .537 | .350 | +.003 | +.000 | +.003 |
| | ConvE | .248 | .521 | .338 | .251 | .521 | .340 | +.003 | +.001 | +.003 |
| | DistMult | .249 | .531 | .342 | .252 | .532 | .345 | +.003 | +.001 | +.003 |
| | HittER* | .268 | .549 | .361 | .271 | .55 | .364 | +.003 | +.001 | +.003 |
| | RESCAL | .263 | .541 | .355 | .265 | .541 | .357 | +.002 | +.001 | +.002 |
| | TransE | .221 | .497 | .312 | .231 | .500 | .320 | +.010 | +.003 | +.008 |
| WN18RR | ComplEx | .438 | .547 | .475 | .443 | .549 | .479 | +.005 | +.002 | +.004 |
| | ConvE | .411 | .505 | .442 | .411 | .505 | .442 | +.000 | +.000 | +.000 |
| | DistMult | .414 | .531 | .452 | .414 | .531 | .452 | +.001 | +.001 | +.000 |
| | HittER* | .437 | .531 | .469 | .44 | .533 | .472 | +.003 | +.002 | +.003 |
| | RESCAL | .439 | .517 | .467 | .44 | .518 | .467 | +.001 | +.000 | +.000 |
| | TransE | .053 | .52 | .228 | .072 | .527 | .242 | +.019 | +.007 | +.014 |
| CoDEx-M | ComplEx | .262 | .476 | .337 | .263 | .476 | .336 | +.001 | +.000 | +.001 |
| | ConvE | .239 | .464 | .318 | .245 | .464 | .321 | +.006 | +.001 | +.003 |
| | HittER* | .262 | .486 | .339 | .266 | .486 | .342 | +.005 | +.000 | +.003 |
| | RESCAL | .244 | .456 | .317 | .248 | .456 | .319 | +.003 | +.000 | +.002 |
| | TransE | .223 | .454 | .303 | .236 | .455 | .309 | +.013 | +.001 | +.006 |
| | TuckER | .259 | .458 | .328 | .26 | .458 | .328 | +.002 | +.000 | +.000 |

Table 7: Removing self-referential prediction from rankings related to completion tasks with irreflexive relations.

We can see that the geometric embedding technique used by TransE seems to be affected much more compared to the other KGE models. If we focus on the other models, we see an improvement of .000 to 0.006 in hits@1. It is interesting to see that hits@10 scores is much less affected. This holds especially for CoDEx-M. We can conclude that in most cases,

where we observe a positive impact of this specific filtering approach, the self-referential prediction is on #1 and the correct alternative is on #2. A number as 0.005 means (for example, HittER* on CoDEx-M), that we observe this pattern in 1 out of 200 completion tasks. This specific filtering technique is thus, much weaker, compared to the filtering that is based on the top-100 AnyBURL ranking.

## Appendix C. Additional Rule Types

AnyBURL can mine more general rules than the rules shown in Section 3 by replacing the constant in the body by an unbound variable. Some examples are listed in the following.

$$gender(X, female) \leftarrow profession(X, A) \tag{8}$$

$$citizen(X, UK) \leftarrow bornIn(X, A) \tag{9}$$

Rule (8) specifies the probability that something that has a profession (no matter what profession) is female. Note that there is also a rule with the same body and the head $gender(X, male)$. Such rules specify a value distribution for a certain type of entities. They have in most cases a rather low confidence and become relevant if nothing else is known that can be understood as a stronger signal.

The final type of rules is a new type that has been added to the latest version of AnyBURL recently. This rule type is based on the fact that asking the knowledge base completion task $r(a, ?)$ implies that there is a correct answer to ?. The confidence of such a rule specifies the probability that a randomly chosen $r$-triple has a specific value in its subject or object position without considering any other information than the distribution of values. Here are some examples for these rules.

$$gender(X, female) \leftarrow () \tag{10}$$

$$citizen(X, UK) \leftarrow citizen(X, A) \tag{11}$$

Without these rules, AnyBURL would sometimes create an empty candidate ranking. These rules yield a kind of default answer to avoid such cases. However, in some situations these rules have an influence that is too strong. Thus, their confidence is multiplied by 0.1 (for Rule (8) and (9)) and 0.01 (Rule (10) and (11)). Without this modifications important entities, e.g., USA, are too often ranked high without any specific reason. This setting is currently the default setting of AnyBURL, which is constantly used across all datasets.

## Appendix D. Summary Statistics and Datasets

| Dataset | #Entities | #Relations | #Triples | | |
| --- | --- | --- | --- | --- | --- |
| | | | Train | Valid | Test |
| FB15k-237 | 14 505 | 237 | 272 115 | 17 535 | 20 466 |
| WNRR | 40 559 | 11 | 86 835 | 3 034 | 3 134 |
| CoDEx-S | 2 034 | 42 | 32 888 | 1 827 | 1 828 |
| CoDEx-M | 17 050 | 51 | 185 584 | 10 310 | 10 311 |
| CoDEx-L | 77 951 | 69 | 551 193 | 30 622 | 30 622 |

Table 8: Datasets and summary statistics.

## Appendix E. Normalizing KGE Scores

As KGE scores can be scattered over a wide value range, we need to normalize the score before we compute the aggregated score. We could, for example, map the score of the #1 rank to 1.0 and the score of the #k rank to 0. The problem of this approach becomes clear when looking at the completion tasks *locatedIn(?,US)* and *locatedIn(?,Liechtenstein)*. If we set $k$ to 100 (the parameter setting we choose in our experiments) we will probably have a situation where the first completion task might yield reasonable candidates within the top-k ranking. For the second ranking this might not be the case, and only a few top candidates are correct. Nevertheless, we would map the candidate at position #100 in both cases on 0 and in the US case a significantly higher value would be more appropriate. Instead of that, we map the KGE score to $[min, max]$ with $max = conf(\mathcal{A}[1], r(e, ?))$ and $min = conf(\mathcal{A}[k], r(e, ?))$, i.e., we map to a range that is defined by the confidence of first and last candidate in the AnyBURL ranking. We are aware that there might be more sophisticated techniques, however, for our purpose this approach turned out to work quite well.

We abstained from other approaches that defines the mapping on a global level by looking at all possible scores that can be derived from the embedding space. Such an approach has to solve the inherent problem that there is probably a wide range of scores that belong to clearly wrong triples.

## Appendix F. Further Results Related to the Ablation Study

In the following Tables, we present the best relation specific MRR results and $\beta_{r,ht}$ values in different settings. Tables 9 and 11 list the values learned with the default setting that we used to achieve the main results shown in Figure 1 for CoDEx-L for the models ComplEx and HittER*, respectively. In general, the values for $\beta_{r,ht}$ vary between the different values of the search space demonstrating the complementary nature of the two approaches (latent vs. symbolic).

In Tables 10 and 12 we report values that were learned in a setting where the search space for $\beta$ is restricted to $\{0.0, 1.0\}$. This mimics a behaviour where a hard selection between the two approaches must be made for each direction/relation in the dataset. For ComplEx in Table 10, the $\beta$ values are almost equal distributed whereas for HittER* most of the time $\beta$ is selected to be 0. A possible reason is the fact that we show these results for the most frequent relations. We found a more equally distributed pattern when considering all the relations of CoDEx-L.

| | head direction | | tail direction | |
| relation $r$ | best MRR | $\beta_{r,h}$ | best MRR | $\beta_{r,t}$ |
| --- | --- | --- | --- | --- |
| occupation | .015 | .20 | .549 | .10 |
| country of citizenship | .040 | .10 | .847 | .20 |
| languages spoken | .022 | .15 | .904 | .25 |
| place of birth | .037 | 1.0 | .316 | .55 |
| educated at | .044 | .20 | .300 | .20 |
| genre | .044 | .20 | .455 | .15 |
| cast member | .093 | .35 | .063 | .30 |
| place of death | .053 | .45 | .446 | .35 |
| member of | .115 | .15 | .509 | .20 |
| member of political party | .042 | .10 | .630 | .25 |

Table 9: The best relation specific MRR values on the valid set and the corresponding $\beta_{r,ht}$ values for ComplEx on CoDEx-L for the 10 most frequent relations.

| relation $r$ | head direction | | tail direction | |
|---|---|---|---|---|
| | best MRR | $\beta_{r,h}$ | best MRR | $\beta_{r,t}$ |
| occupation | .013 | 1.0 | .535 | 0.0 |
| country of citizenship | .038 | 0.0 | .828 | 0.0 |
| languages spoken | .019 | 0.0 | .828 | 0.0 |
| place of birth | .037 | 1.0 | .316 | 1.0 |
| educated at | .043 | 0.0 | .276 | 1.0 |
| genre | .039 | 0.0 | .425 | 0.0 |
| cast member | .091 | 1.0 | .060 | 1.0 |
| place of death | .046 | 1.0 | .440 | 1.0 |
| member of | .099 | 0.0 | .472 | 0.0 |
| member of political party | .038 | 0.0 | .590 | 0.0 |

Table 10: The best relation specific MRR values on the valid set and the corresponding $\beta_{r,ht}$ values for ComplEx on CoDEx-L for the 10 most frequent relations when beta is restricted to be in {0,1}.

| relation $r$ | head direction | | tail direction | |
|---|---|---|---|---|
| | best MRR | $\beta_{r,h}$ | best MRR | $\beta_{r,t}$ |
| occupation | .016 | .10 | .567 | .10 |
| country of citizenship | .047 | .05 | .859 | .20 |
| languages spoken | .025 | .05 | .919 | .20 |
| place of birth | .038 | .90 | .327 | .40 |
| educated at | .045 | .15 | .318 | .10 |
| genre | .046 | .05 | .469 | .25 |
| cast member | .099 | .20 | .067 | .20 |
| place of death | .057 | .30 | .468 | .20 |
| member of | .124 | .15 | .546 | .10 |
| member of political party | .044 | .00 | .641 | .20 |

Table 11: The best relation specific MRR values on the valid set and the corresponding $\beta_{r,ht}$ values for HittER* on CoDEx-L for the 10 most frequent relations.

| relation $r$ | head direction | | tail direction | |
|---|---|---|---|---|
| | best MRR | $\beta_{r,h}$ | best MRR | $\beta_{r,t}$ |
| occupation | .016 | 0.0 | .560 | 0.0 |
| country of citizenship | .047 | 0.0 | .839 | 0.0 |
| languages spoken | .023 | 0.0 | .914 | 0.0 |
| place of birth | .037 | 1.0 | .310 | 0.0 |
| educated at | .041 | 0.0 | .305 | 0.0 |
| genre | .045 | 0.0 | .442 | 0.0 |
| cast member | .091 | 1.0 | .061 | 0.0 |
| place of death | .049 | 0.0 | .447 | 0.0 |
| member of | .122 | 0.0 | .532 | 0.0 |
| member of political party | .044 | 0.0 | .611 | 0.0 |

Table 12: The best relation specific MRR values on the valid set and the corresponding $beta_{r,ht}$ values for HittER* on CoDEx-L for the 10 most frequent relations when $\beta$ is restricted to be in {0,1}.