# OpenReview forum: "Why a Naive Way to Combine Symbolic and Latent Knowledge Base Completion Works Surprisingly Well"
_AKBC.ws/2021/Conference — AKBC 2021_

### Official Review · Reviewer_Bzsz · 2021-07-12
**Well motivated ensemble of symbolic and soft KB completion models.**

**Rating:** 7
**Confidence:** 3

**Review:**

The authors investigate ways to ensemble symbolic and soft KB completion models. The symbolic approach infers rules from the training data, and the soft approach learns to embed the KB into a geometric space.
The authors start by analyzing the behavior of the two approaches and specify cases where each of them fail.
This analysis is used to come up with several heuristics for combining the predictions of the two approaches.
On several datasets, the authors show that the proposed ensemble is better (although not drastically) than the two approaches on their own.

The paper is well motivated, and can give insights about the task and its leading models. The paper is also well written and easy to follow.

Having said that, the improvements the paper shows are somewhat limited (few MRR points), and ensembling two different approaches together is standard for pushing the performance.

---

> ### Author Response · Authors · 2021-07-26
> **Thank you for your feedback**
>
> We are happy about the positive impression that the paper made and thank the reviewer for the valuable comments.
>
> Two critical points have been raised, that we address in the following:
>
> 1. *the improvements the paper shows are somewhat limited (few MRR points)*
> 2. *ensembling two different approaches together is standard for pushing the performance.*
>
> (1) The improvements of our aggregation methods are indeed within the scope of a few MRR points for many of the presented results. However, the results for the respective models before using our aggregation method are already state-of-the-art or close to the state-of-the-art. A few MRR points is a significant improvement when starting from an already strong level. At the same time, our approach strongly improves models that perform notably worse such that the resulting ranking is competitive with the best performing models. For instance, quite surprisingly, the aggregation method improves TransE on WNRR from 0.228 to 0.503 which is only 0.003 percentage points less than the best performing model and 0.006 better than AnyBURL.
>
> As a similar point has been raised by Reviewer 6Wgd, we also refer to the answer we give there.
>
> (2) This point is also true in general. However, instead of constructing an ensemble from two arbitrarily chosen models, we investigate the complementary nature of latent representations and a model that uses symbolic representations in terms of rules. Moreover, each candidate that appears in the final ranking has been proposed by the symbolic method. This means that the result of the joint model can still be explained in terms of the rules that generated this candidate. Thus, we keep most of the explanatory power of the symbolic method while exploiting the benefit of the better aggregation of the embedding based method. In that sense, our method is more specific than a standard ensemble. From a different point of view, our work can also be understood as a technique to explain the results of an embedding based KGE model. We tried to make this point a bit clearer by adding some additional sentences related to this aspect (e.g., last sentences of the introduction, last bullet point in the conclusion).

---

### Official Review · Reviewer_hMxP · 2021-07-22
**An interesting paper with well-motivated examples and a simple aggregation approach**

**Rating:** 6
**Confidence:** 3

**Review:**

The paper presents a comparative study on symbolic methods and embedding methods for knowledge graph completion. From the manually observed ranking results of two kinds of approaches, the authors find that rule-based method AnyBURL tends to perform well on complicated rules that express transitivity of multiple relations, and that KG embedding methods sometimes will output meaningless result due to its learned similarity measure, which may be remedied by rule-based methods. Accordingly, a simple aggregation reranking approach is also proposed, and it shows promising performance gain for several embedding approaches in the KG completion task.

I find qualitative analysis in comparing symbolic and embedding-based methods is insightful. The illustrated examples are closely related to the proposed approach, which is simple yet shown to be effective in boosting prediction performance. In general, the paper is well organized and easy to follow. Besides, I still have some questions regarding the analysis and experiments.

- It is intuitive to examine the cherry-picked examples and discover the potential patterns of various methods, but from the macro perspective, it would be better to provide more global statistical information, for instance, how many examples are there similar to the Darwin example? If it accounts for a very large proportion, it is not surprising the proposed method can work well. If there are only very limited similar cases, the performance gain may be caused by other reasons. In other words, larger-scale of analysis is preferred to demonstrate global statistical information.

- Since KG embedding methods can be also used to predict head entity or relation, is the proposed aggregation method able to be applied to these two tasks as well?

- It seems that the coefficient beta in the scoring function plays an important role in balancing the contribution of rule-based confidence scores and embedding-based scores. It would be more convincing to show the ablation study on this parameter.

---

> ### Author Response · Authors · 2021-07-26
> **Thank you for your feedback**
>
> We thank the reviewer for the valuable comments. The following critical points have been raised.
>
> **Reviewer Comment**
>
> 1. *It is intuitive to examine the cherry-picked examples and discover the potential patterns of various methods, but from the macro perspective, it would be better to provide more global statistical information, for instance, how many examples are there similar to the Darwin example? If it accounts for a very large proportion, it is not surprising the proposed method can work well. If there are only very limited similar cases, the performance gain may be caused by other reasons. In other words, larger-scale of analysis is preferred to demonstrate global statistical information.*
>
> **Our answer:**
>
> We agree that a statistical overview would be helpful to estimate the impact of our method, however, it is not easy at or might even be impossible to define an automated procedure to generate these numbers. Understanding a small example as the Darwin example is in itself not trivial at all. It requires to check the complete neighbourhood of the involved entities, but even more important, in our analysis we used common sense knowledge about the types of the entities (some entities are territories, some are countries that can be divided into territories, and so on). Without this knowledge, it would not be possible to understand that these types, which are implicitly encoded in sets of triples, make certain entities “similar” to each other. This is another reason, why an automated statistical analysis is beyond the goal that we can achieve with the paper.
>
> What we checked with all relevant statistics in Table 4 in the Appendix is the impact of using the rule-based ranking as a filter (by setting beta to 0). This is one aspect of an automated method and it helped us to understand that ¼ to ½ of the positive impact can be explained in terms of filtering.
>
> We also analysed the impact of suppressing self-referential predictions as we found this type of error/problem in some of the examples. Here we could provide a full statistical analysis as this type of error can be automatically detected. Results are shown in Table 6 in Appendix B.
>
> However, we are aware that we cannot exactly quantify exactly which type of specific behaviour appears how frequently for each of the observed phenomena.
>
> **Reviewer Comment**
>
> 2. *Since KG embedding methods can be also used to predict head entity or relation, is the proposed aggregation method able to be applied to these two tasks as well?*
>
> **Our answer:**
>
> Indeed, the aggregation method can be applied to others than the tail direction. In fact, the main results in the paper are based on the joint MRR in both, head, and tail directions. Now we clarified this in the introduction of the updated paper version
>
> We abstained from evaluating the model towards answering queries about relations to be consistent with the common evaluation protocols in order to make our results comparable to and usable for a wide research community. Nevertheless, we acknowledge that this would be an interesting direction.
>
> **Reviewer Comment**
>
> 3. *It seems that the coefficient beta in the scoring function plays an important role in balancing the contribution of rule-based confidence scores and embedding-based scores. It would be more convincing to show the ablation study on this parameter.*
>
> **Our answer:**
>
> We agree with the reviewer in that a more detailed ablation study regarding the hyperparameter beta would benefit the work. For that reason, we added a new Subsection 6.3 where we focussed on several experiments reporting about results for fixed beta-values and results for restricted sets of possible values. Overall, we added five new tables to the paper.
>
> Note that the first part of the new section contains material we put in the previous version of the paper at the end of Section 6.2. However, with the new structure it fits better at the beginning of Section 6.3. The second part of Section 6.3 contains new material. This also holds for Table 3 (in the main text) and Table 9 to 12 in the Appendix. We believe the new ablation material improved the paper and thank the reviewer for the comment.

---

### Official Review · Reviewer_6Wgd · 2021-07-22
**Some insights, but only minor improvements obtained.**

**Rating:** 6
**Confidence:** 4

**Review:**

Knowledge Graph Embedding (KGE) methods solve link prediction problems by learning node embeddings and a score function such that the correct triples are ranked the highest. On the hand, AnyBURL is a symbolic alternative which learns rules for ranking triples. The paper proposes a simple aggregation approach for combining the ranking results from KGE methods with those from AnyBURL. The motivation of this approach is to ultimately harness the strengths of both symbolic and sub-symbolic methods. The method is evaluated by studying the improvement in link prediction performance using benchmark datasets.

Overall, the idea presented in the paper is interesting because of its simplicity. Despite the weak results obtained in the empirical study, I think it presents an exciting new direction for the integration of symbolic and sub-symbolic methods. However, I strongly advice the authors to improve the text to make it read better.

Strengths:
* The aggregation function (in Section 5) is simple and the intuition behind this approach is easy to understand.
* Structure of the paper is easy to follow.
* The analysis presented in Section 4.2 are insightful.
* The results from the rule-based approach are explainable. However, this feature has not been thoroughly studied.

Weaknesses:
* The writing style of the paper could be improved to make it read better. There are many awkwardly phrased sentences throughout the text: “We abstain here from...“; “Unfortunately, AnyBURL is to our knowledge the only symbolic approach...“; “Within this work we have not presented a sophisticated new knowledge base completion method.”
* The results reported in the paper show only very little improvements (1-3% improvement in Mean Reciprocal Rank). Therefore, I feel that the title of the paper is misleading.

Minor Typos:
Section 4.2 (page 5) “We first look at two predictions” → “three predictions”
Abstract (page 1) “embbedings” → “embeddings”
Related Work (page 2) “the the” → “the”
Section 6.1 (page 8) “Further” → “Furthermore”

---

> ### Author Response · Authors · 2021-07-26
> **Thank you for your feedback**
>
> First, we thank the reviewer for the valuable comments.
>
> We thank the reviewer for pointing to problems related to the readability of the paper. We rephrased all the mentioned passages and fixed all mentioned typos. We also plan to check the paper with the help of an external person that was not involved in the writing.
>
> Aside from readability and typos the following point has been raised:
>
> **Reviewer Comment**
>
> *The results reported in the paper show only very little improvements (1-3% improvement in Mean Reciprocal Rank). Therefore, I feel that the title of the paper is misleading.*
>
> **Our Answer**
>
> We measured the improvements in terms of percentage points, and we made this more explicit in the revised version. For example, ComplEx improved its MRR on CoDEx-L from 29.4% to 32.9% which is a plus of 3.5 percentage points. The improvement in terms of percentage points sounds small, however, this is an improvement of 11.9 percent. We added HittER* to our experiments to ensure that the improvements can also be observed for the latest state-of-the-art. Here we have an improvement on CoDEx-L from 32.2% to 34%, which is a plus of 1.8 percentage points and an improvement of 5.5 percent. Please also note that the average improvement over all datasets and models is 2.6 percentage points if we exclude TransE (including TransE would increase the average improvement to 4.24 percentage points) which is quite substantial.
>
> There are many research publications that report on small incremental improvements based on complex architectures, complicated theoretical frameworks, or a specific modification within these architectures. Within our experiments, we observed constant and significant improvements over all models and datasets that we used. We found the consistency of these results noteworthy (maybe even surprising) given that our method is very simple (almost naive).

---

### Author Response · Authors · 2021-07-30
**Thank you**

Thank you for your comments! Based on these comments, we uploaded a revised version of our paper. The most important improvement is related to a new Section 6.3, which contains an ablation study that analyses the impact of the beta parameter.

Furthermore, we give a specific answer to each reviewer in the comments below.

---

### Decision · Program_Chairs · 2021-08-17

**Decision:**

Accept

**Comment:**

This paper presents an ensemble scheme to combine predictions from embedding and rule-based KG completion methods. While the proposed method is extremely straightforward, the aggregated method does perform consistently better compared to the component methods.

Overall, we think the paper is interesting, well-motivated, and adds value to the AKBC program. Hence, we recommend acceptance.